# OpenReview forum: "Towards Understanding Subliminal Learning: When and How Hidden Biases Transfer"
_ICLR.cc/2026/Conference — ICLR 2026 Poster_

### Official Review · Reviewer_xtku · 2025-10-24

**Soundness:** 4
**Presentation:** 4
**Contribution:** 3
**Rating:** 8
**Confidence:** 3

**Summary:**

The authors investigate the phenomenon of subliminal learning through a mechanistic lens, asking under which conditions it turns up and which parts of the network are primarily responsible for this behaviour, and whether it is transferable between different models. More generally, they show that this is a rather fragile phenomenon. While it may occur even for updating only parts of the network, it is rather easily avoided by changing prompts or slightly modifying the data. Overall, this work gives more insights into a security question in practical large language models.

**Strengths:**

- The work investigates the underlying reasons for subliminal learning in language models in a very structured, easy-to-follow manner.
- It successfully refutes the observation from dataset distillation that the full distribution of the labels is necessary for learning, showing that the discrete tokens sampled greedily are enough to learn subliminally.
- They introduce the concept of divergence tokens, which are the tokens that differ in prediction between teachers with different hidden traits.
- The location of the origin of the subliminal learning seems to lie within the early layers of the network.
- While the main focus is on understanding the mechanism, the work shows that the phenomenon is fragile and can be quite easily avoided using simple and cheap strategies.

**Weaknesses:**

See questions.

- The code is not yet provided; therefore, its reproducibility cannot be assessed.

**Questions:**

- Why do you define divergence tokens via a counterfactual teacher (CF), rather than a non-finetuned teacher (NFT)? Intuitively, divergence tokens defined between the original model (OM) and the NFT should naturally have a higher influence than the same tokens, since their contribution to the cross-entropy loss should be bigger during training. Could you clarify the reason for using the CF instead?
- The setting is kept identical with the previous work. To assess stability, it would have been useful to investigate further noun groups, as well as investigate how "far" different tokens can be from each other semantically to allow for this learning (can one transfer a model's favourite animal being a ship more easily or less easily than a monkey?)

Hypothetical curiosity questions, feel free to ignore this:
- You observe that often you do not have subliminal learning, and you avoid it easily. Yet you observe singular cases where the models do transfer some traits. Do you have an intuition about what properties could be predictive of the transferability? Or otherwise, how statistically significant is the transfer (how many pairs did you test, to arrive at some transfer for some samples)?
- Does the analysis also hold for truly memorised knowledge, like passwords or emails, that are made out of character sequences and not single tokens?
- It could be interesting to engage mechanistically with the previous work's hypothesis, that the gradients are pushing the student into the teacher's direction. By comparing the models in weight-space, is this in line with the hypotheses from this work, e.g. the model moves closer to the teacher in weight space, especially in those early layers?

---

> ### Author Response · Authors · 2025-11-20
>
> We thank the reviewer for their thorough and constructive feedback. Below, we address remaining concerns and questions.
>
> > Code is not yet provided
>
> We have added pseudocode for divergence token identification (Figure 10, Appendix D), as suggested by reviewer bqv2. We’ll release code upon acceptance. However, if the reviewer requires code to make their final decision, we’re happy to provide it.
>
> > Why do you define divergence tokens via a counterfactual teacher (CF), rather than a non-finetuned teacher (NFT)?
>
> We initially experimented with using the non-finetuned teacher (NFT). However, we found that the bias-inducing system prompt shifted prediction so strongly that almost every token differed from the biased teacher. In contrast, counterfactual teachers, sharing a similar system prompt, produce more aligned predictions. This likely reflects the strong effect of system prompts on LLM outputs (see, e.g., [1]).
>
> > It would have been useful to investigate further noun groups
>
> We added results on tree preferences; see Figure 16 (Appendix G). The results are overall consistent with the other results.
>
> > Can one transfer a model's favourite animal being a ship more easily or less easily than a monkey?
>
> As suggested, we tried several non-animal concepts (ship, piano, airplane); see Figure 37 (Appendix Q). Subliminal learning mostly does not work here. However, we surprisingly found a small bias transfer of ~1.5% for piano.
>
> > Do you have an intuition about what properties could be predictive of the transferability? Or otherwise, how statistically significant is the transfer?
>
> We suspect this comment refers to the cross-model mixing experiments (Figure 7). If that is not the case, please let us know.
>
> We tested only animal-model pairs for which both models individually showed subliminal learning for the animal, see Figure 13 (base & FT bars). This resulted in the three animals used for this experiment. We found transfer in 2/6 cases (and small levels of transfer for another 2). Cloud et al. [2] suggested shared initializations. However, also similar training data (with intentional or inadvertent cross-model data distillation) could explain cross-model transfer. That being said, we’d leave further controlled analysis, including more biases and model pairs, to future work.
>
> > Does the analysis also hold for truly memorised knowledge, like passwords or emails, that are made out of character sequences and not single tokens?
>
> This is an interesting question. Note that we’ve already included experiments for misalignment transmissions (i.e., behavioral changes) and some animals consist of multiple tokens. We’d however leave the suggested settings for future work.
>
> > Gradients are pushing the student into the teacher's direction. [...] The model moves closer to the teacher in weight space, especially in those early layers?
>
> Studying if and how the student moves closer to the teacher in weight space could provide interesting insights which likely complement the theoretical results of Cloud et al. [2]. While we have not studied weight space similarities, we have conducted some preliminary experiments studying activation similarities between finetuned students and their biased teachers (compared to the base models vs. biased teachers). However, the analysis has not yielded conclusive results so far.
>
> ---
>
> [1] Neumann et al. "Position is Power: System Prompts as a Mechanism of Bias in Large Language Models (LLMs)." FAccT 2025.
>
> [2] Cloud et al. "Subliminal learning: Language models transmit behavioral traits via hidden signals in data." arXiv 2025.

---

### Official Review · Reviewer_VLoB · 2025-10-29

**Soundness:** 2
**Presentation:** 3
**Contribution:** 2
**Rating:** 4
**Confidence:** 4

**Summary:**

The paper proposes an explanation for the subliminal learning phenomenon based on divergence tokens---specific tokens in generated sequences where teacher models with different biases begin to diverge in their predictions. The authors show that these tokens alone are sufficient for bias transfer when fine-tuning a student model on the teacher-generated data, while masking them out suppresses the effect. They further demonstrate that fine-tuning only the early layers of the student model is enough to transfer the teacher’s bias. Finally, they observe that small prompt rephrasings (even when generated by the biased teacher) or mixing data from several teachers with different biases reduce bias transmission.

**Strengths:**

- Attempts to analyze a timely and relevant phenomenon using controlled settings

- The experiments showing where "divergence tokens" affect finetuning provide causal evidence

- Empirical insights into which layers matter for bias transfer when finetuning on data generated by a biased teacher. This is a nice contribution.

- The paper presents a potential "defense" for bias transfer in subliminal learning, which includes using a multi-teacher setting.

**Weaknesses:**

- Main concern: While the experiments are well conducted, the conceptual depth of the contribution is limited. The identification of “divergence tokens” is somewhat tautological---by definition, these are the tokens where biased teachers diverge, so it is expected that these tokens correlate with bias transfer (assuming we know subliminal learning happens). The paper convincingly shows that these tokens are enough for transmitting bias, but it does not explore why they carry this information or what within their representations enables the effect. A more mechanistic explanation would greatly strengthen the claim that divergence tokens encode teacher bias rather than simply marking where model behaviors differ.

- The filtering step in Section 5.1 introduces ambiguity: by excluding non-argmax tokens, the authors effectively bias the “non-greedy” dataset toward greedy decoding

- The relationship between divergence tokens and entangled tokens also remains underspecified. The authors assert that entanglement is not required, but do not empirically quantify or visualize the overlap between the two sets.

**Questions:**

- In case I'm missing something, please explain why the fact that divergent tokens lead to divergent learned behavior in subliminal learning is not trivial

- If you have any evidence for the information encoded by divergence tokens, please share it. What are potential ways to tackle this question?

- Are divergence tokens semantically or representationally different from the entangled tokens?

- Please clarify the procedure of filtering data in 5.1. Doesn't filtering non-greedy data in Sec. 5.1 effectively make the dataset greedy?

- How consistent are divergence tokens across random seeds or prompt variants?


Nitpick: L047: double “through”

---

> ### Author Response · Authors · 2025-11-20
>
> We thank the reviewer for taking the time to review and the constructive feedback. We address the remaining concerns and questions below.
>
> > Identification of “divergence tokens” is somewhat tautological---by definition, these are the tokens where biased teachers diverge, so it is expected that these tokens correlate with bias transfer (assuming we know subliminal learning happens); why the fact that divergent tokens lead to divergent learned behavior in subliminal learning is not trivial
>
> We agree that divergence tokens are positions where biased teachers produce different predictions for the same prefix. However, we disagree that it’s “trivial” that these tokens *cause* bias transfer in subliminal learning:
>
> The *literature offers other hypotheses not relying on teacher disagreements*, such as token entanglement (“special numbers”) or logit leakage via random sampling of non-argmax tokens. They attribute subliminal learning to specific token-level semantics or statistical logit leakage. While they seem very intuitive, we found that they are *not required* (see Section 4). The divergence token hypothesis provides a different and novel explanation (Section 5 & Appendix C).
>
> We’d also like to add more nuance to the reviewer’s comment that “it is expected that these [divergence] tokens correlate with bias transfer”. We believe the following insights go meaningfully beyond what’s expected: For example, we show that it is typically not sufficient that divergence tokens merely appear in the input context when predicting a non-divergence token (w/o-div-token bars in Figure 3), whereas transfer reliably occurs when divergence tokens are directly predicted and gradients are computed on them (div-token bars). This *clarifies the role of divergence tokens more precisely*. Notably, divergence tokens also only make up, e.g., ~5% of training tokens! In contrast, training on the other, e.g., 95%, tokens typically doesn’t transfer biases. This shows that the *subliminal learning signal is surprisingly highly concentrated in few tokens*. Moreover, it is not obvious whether there’s a *complementary role between divergence and non-divergence tokens*, e.g., non-divergence tokens carrying necessary complementary signals. However, Figure 3 suggests that divergence tokens alone account for nearly the entire subliminal learning phenomenon.
>
> Finally, we’d like to clarify that *identifying divergence tokens in practice is not as simple as our presentation may have made it seem*. For example, see the discussion with reviewer xtku about the choice of counterfactual teachers.
>
> > If you have any evidence for the information [(i.e., teacher bias)] encoded by divergence tokens, please share it. What are potential ways to tackle this question?
>
> We probed the hidden activations at divergence token positions using sparse autoencoders (SAEs) but didn’t find interpretable token-level information. This may reflect limitations of the approach for this analysis. However, it could also suggest that subliminal learning may not work via token-level semantics but instead relies on an accumulated signal, i.e., *only multiple divergence tokens together, possibly over multiple samples, carry the hidden teacher bias*. The intuition for this mechanism is described in Section 5 and more detailed in Appendix C.
>
> > The authors assert that entanglement is not required, but do not empirically quantify or visualize the overlap between the two sets.; Are divergence tokens semantically or representationally different from the entangled tokens?
>
> The added Table 2 (Appendix H) shows that the overlap is indistinguishable from what would be expected by random chance, indicating that the two concepts are indeed distinct. Besides that, note the conceptual difference: entangled tokens are defined globally (across all prompt-completion pairs), whereas divergence tokens are instance-specific, occurring only under specific prefix-teacher combinations.
> > Doesn't filtering non-greedy data in Sec. 5.1 effectively make the dataset greedy?
>
> We apologize for the confusion. In the temperature-sampling settings, we classify a token as a divergence token iff it satisfies Definition 5.1 *and* is the teacher’s argmax. Tokens that satisfy Definition 5.1 but are not argmax (due to sampling) are treated as non-divergence tokens; importantly they aren’t removed. Thus, the dataset remains non-greedy. We have clarified this in the revision.
>
> > How consistent are divergence tokens across random seeds or prompt variants?
>
> Divergence tokens are prefix-dependent and therefore sensitive to prompt paraphrases, as can be seen in Figure 6. For greedy decoding, divergence tokens are deterministic and consistent across seeds. Under stochastic sampling, they naturally vary across seeds due to different sampled token trajectories.

---

### Official Review · Reviewer_bqv2 · 2025-10-31

**Soundness:** 2
**Presentation:** 3
**Contribution:** 2
**Rating:** 4
**Confidence:** 4

**Summary:**

The paper studies subliminal learning—hidden trait transfer during distillation on task-unrelated data—and asks when/why it happens. The main empirical claim is that transfer does not require logit leakage or “global token entanglement.” Instead, the effect concentrates on a small set of divergence tokens where biased teachers would deterministically choose different next tokens under the same prefix. Masking loss on those tokens suppresses transfer; training only on them often preserves or amplifies it. A second result is that early layers are disproportionately causal (one early layer often suffices to induce transfer), and a third is fragility: minor prompt paraphrases or mixing teachers weakens the effect. The work spans Qwen2.5-7B-Instruct and Gemma-3-4B-it with both greedy and temperature sampling; it reports multiple ablations (entangled-token filtering, divergence-only loss, single-layer LoRA, paraphrase/mix controls).

**Strengths:**

Greedy generation + explicit removal of entangled tokens rules out two popular explanations and isolates the phenomenon.

Divergence-token causal handle. The mask-in/mask-out experiments are decisive and easy to replicate; they also give a knob to control transfer.

Layer localization with intervention. Early layers matter (and sometimes suffice). This is a crisp, actionable take-home for defense/mitigation.

Honest about artifacts/exceptions. The Qwen system-prompt name leak and “penguin” exceptions are called out rather than buried.

**Weaknesses:**

Scope is narrow. The main demonstrations use animal-preference transfer on number-list tasks. There is one misalignment transfer (financial risk advice), but the bulk is a single stylized setup; stronger claims would need more domains (code with unit tests, factual QA, stylistic tics) and non-numeric outputs.

Detection pipeline details could be crisper. The exact procedure to identify divergence tokens (across which counterfactuals? earliest only or all? thresholding?) is scattered; a short algorithm box with counts per setting would help reproducibility.

Evaluation bleed-through. The numbers task performance is measured on training prompts; test-set paraphrases are used elsewhere, but a fully held-out task set for the performance axis would remove an avoidable caveat.

Interplay with sampling and decoding at test time. Most claims are made with specific decoding choices (greedy/temperature=1). It would be useful to show whether changing decoding at evaluation (e.g., nucleus, different temps) weakens or strengthens measured transfer.

Theory is intuitive, not predictive. The Appendix C picture is helpful but doesn’t yet predict which prefixes will diverge or how many you need to see a given effect size; a simple probabilistic model calibrated to data would raise the ceiling.

**Questions:**

Operationalizing divergence detection. Do you take all positions where any counterfactual teacher’s argmax differs from the factual teacher on the same prefix, or only the first such position per sequence? If multiple, how do you weight them?

Counting & impact. What fraction of tokens are divergence tokens per model/sampler (exact numbers, not only ranges), and how does transfer scale with (i) #divergence tokens seen in training and (ii) their positional depth?

Generalization beyond animals. Can you replicate the core Figure-3 result on a non-numeric, non-preference trait (e.g., writing style, friendliness) where evaluation isn’t a single word?

Mitigations at training time. Instead of removing divergence tokens, have you tried down-weighting them or mixing unbiased/other-teacher data only at divergence positions? This could retain task performance while blunting transfer.

Early-layer interventions. Your causal/patching analysis suggests early layers are decisive. If one wanted to prevent transfer, would freezing (or spectral-norm clipping) just the first N layers blunt it while preserving task learning?

Judge for leakage at inference. When you paraphrase prompts (Figure 6), transfer drops while task performance stays high. Is that because divergence tokens shift to later positions or become rarer? Any shifts in the distribution of divergence positions under paraphrase?

---

> ### Author Response · Authors · 2025-11-20
>
> We thank the reviewer for the thorough review and the constructive criticisms. Below, we address the remaining points below.
>
> > Stronger claims would need more domains; Can you replicate the core Figure-3 result on a non-numeric, non-preference trait (e.g., writing style, friendliness) where evaluation isn’t a single word?
>
> We tried to replicate the misalignment transmission via chain-of-thought experiment from Cloud et al. [1], using open-source models instead of closed-source ones. This replication is needed as a prerequisite for our masking experiments in Figure 3. So far, we were unable to reproduce their setting. We have thus contacted the authors, who replied that they haven’t done open-source models in that setting yet and currently don’t plan to do so. Thus, we cannot foresee if we can make the base setting work and therefore hope for the reviewer’s understanding that without it, replicating Figure 3 results in that setting is not possible at this time.
>
> Note that we added results on tree preferences (Appendix G) and that evaluation of misalignment is not a single word.
>
> > Detection pipeline details could be crisper. [...] across which counterfactuals? earliest only or all? thresholding? [...] Do you take all positions where any counterfactual teacher’s argmax differs from the factual teacher on the same prefix, or only the first such position per sequence? If multiple, how do you weight them?
>
> We added pseudocode in Figure 10 (Appendix D). Appendix D also lists the counterfactuals used in each setting. We take all divergence tokens (unless stated otherwise), don’t threshold, and assign equal weight to all divergence tokens.
>
> > Numbers task performance is measured on training prompts
>
> As the task is memorization-based, we therefore used the training set. As suggested, we nonetheless added results on the held-out set (Figure 28, Appendix J).
>
> > It would be useful to show whether changing decoding at evaluation (e.g., nucleus, different temps) weakens or strengthens measured transfer.
>
> We re-evaluated using nucleus sampling and different temperatures. Figures 14 & 15 (Appendix F) show consistent results across sampling parameters.
>
> > Appendix C picture is helpful but doesn’t yet predict which prefixes will diverge or how many you need to see a given effect size; a simple probabilistic model calibrated to data would raise the ceiling.
>
> We now analyze how transfer varies with both the number and positional depth of divergence tokens (Figures 21a–d, Appendix H); in response to the other questions. The results sometimes show unclear patterns, suggesting that fitting a probabilistic model may be non-trivial and requires distinguishing the impact of individual divergence tokens. We see this as a promising future direction but beyond the scope of the present work.
>
> > What fraction of tokens are divergence tokens per model/sampler?
>
> We added Table 1 (Appendix H) and it shows that the fractions of tokens are consistent across the biases. We also added the fractions for the tree preference (Appendix G) and misalignment experiments (Appendix M).
>
> > How does transfer scale with (i) #divergence tokens seen in training and (ii) their positional depth? Instead of removing divergence tokens, have you tried down-weighting them or mixing unbiased/other-teacher data only at divergence positions?
>
> Figures 21a & b show that, for Gemma, randomly subsampling divergence tokens steadily reduces bias transfer. For Qwen, the pattern is less clear suggesting that divergence tokens may vary in effect strength. Figures 21c & d do not indicate a relation between positioning of divergence tokens and hidden bias transfer. Figures 21e & f show that mixing in other-teacher data only at divergence positions reduces hidden bias transfer. In fact, often 50% mixing in other data has a similar or even stronger effect than masking. We did not run down-weighting experiments since we expect only weaker effects (masking effectively corresponds to weights 0 & 1). Though, if the reviewer considers it necessary, we’re happy to run this as well.
> > If one wanted to prevent transfer, would freezing (or spectral-norm clipping) just the first N layers blunt it while preserving task learning?
>
> As suggested, we froze different numbers of first layers in Appendix S. Figure 39 shows that freezing the first 10+ layers effectively suppresses subliminal learning and Figures 40 & 41 show that task performance remains nearly intact.
>
> > When you paraphrase prompts (Figure 6), transfer drops while task performance stays high. Is that because divergence tokens shift to later positions or become rarer?
>
> We added Figure 29 (Appendix K). It shows that divergence tokens become less frequent under paraphrases and positional shifts are inconsistent, showing no consistent trend.
>
> ---
>
> [1] Cloud et al. "Subliminal learning: Language models transmit behavioral traits via hidden signals in data." arXiv 2025.

---

> > ### Author Response · Authors · 2025-11-28
> > **Re: Official Comment by Authors**
> >
> > We thank the reviewer for their patience and hope that the new experiment now addresses their earlier concern:
> >
> > > Stronger claims would need more domains; Can you replicate the core Figure-3 result on a non-numeric, non-preference trait (e.g., writing style, friendliness) where evaluation isn’t a single word?
> >
> > Since our initial response, we successfully implemented an open-source replication of the Cloud et al. setting work, which in turn allowed us to replicate our other results in this additional domain (see Appendix N). In this new experiment, we transmitted misaligned tendencies using non-preference data and non-numeric prompts, as suggested. Specifically, we transmitted misaligned tendencies through the teacher’s responses to GSM8K’s math word problems. Note that misalignment evaluations are full sentences rather than single words.
> >
> > The results are consistent with our previous misalignment results (Figure 8 and Appendix M).

---

### Author Response · Authors · 2025-11-20
**General response**

We thank the reviewers for their constructive feedback, which helped us improve the manuscript.

The reviews were largely positive. Reviewers highlighted that the work addresses a “timely and relevant phenomenon” (VLoB) “in a very structured, easy-to-follow manner” (xtku). They liked that we “rule[d] out two popular explanations” (bqv2), described the divergence token hypothesis and supporting experiments as “decisive” (bqv2), found the analysis of which layers matter “a nice contribution” (VLoB), and appreciated the finding that subliminal learning “can be quite easily avoided using simple and cheap strategies” (xtku).

In response to the reviews, we improved the manuscript as follows, with changes highlighted in blue in the updated version:

* While running additional experiments suggested by the reviewers, we found a small bug in the w/o-div-tokens experimental setting. We fixed this and re-ran all affected experiments. All conclusions remain unchanged. (Updated the w/o div-tokens bars in Figures 3, 8, the corresponding Appendix figures, and fractions of (non-)div-tokens.)
* Added further details and pseudocode for divergence token identification (Appendix D).
* Added robustness results to different sampling parameters during evaluation (Appendix F).
* Additional experiments with tree preferences (Appendix G).
* Extended analysis of divergence tokens, including their overlap with entangled tokens or the effect of mixing other teacher data only for divergence tokens (Tables 1 & 2, Figure 21).
* Added analysis on the impact of paraphrasing on divergence tokens (Appendix K).
* Additional experiment on transferring misalignment via answers to GSM8K math questions (Appendix N)
* Additional experiments on whether a non-animal preference (e.g., ship) can be turned into the favorite-animal preference (Appendix Q).
* Added results showing that freezing the first N layers can also suppress subliminal learning (Appendix S).
* Corrected typos, made clarifications, etc.

---

### Author Response · Authors · 2025-11-28
**Authors' follow-up**

Dear reviewers,

Thank you again for your constructive feedback, which has helped us significantly strengthen the manuscript (please see our [global response](https://openreview.net/forum?id=IelhmYSjPt&noteId=MLMeuh2NSQ) for the detailed change log).

We hope our responses have addressed your questions and concerns. As the discussion phase is ongoing but already concludes on December 3, we’d appreciate hearing your thoughts and/or any follow-up questions you may have. We’re happy to clarify or expand on anything that would help your final assessment.

Please note that an additional experiment based on the reviewer’s bqv2 suggestion led to few reference updates. We have revised earlier responses to ensure consistency with the updated manuscript.

Thank you for your time and consideration.

---

### Meta-Review · Area_Chair_sXyA · 2026-01-09

**Summary:**

##### Reviewer bqv2

(1) Demonstrations are restricted to one domain; (2) methodological details of the detection pipeline are missing; (3) some experimental choices are insufficiently justified; (4) theory does not enable predictions of many types.

##### Reviewer VLoB

(1) The identification of divergence tokens as the cause of bias transfer during hard distillation is not a mechanistic explanation; (2) concern with small methodological details; (3) the relationship between divergence tokens and entangled tokens is not clear.

##### Reviewer xtku

None significant.

**Reviewer Concerns:**

The concerns of Reviewer bqv2 seem to be addressed by (1) the additional replication in an additional domain in Appendix N and (2–4) by written justification. The concerns of Reviewer VLoB seem to be addressed by more thorough explanation (2, 3), and for (1) by a detailing of exactly what is meant by mechanistic in the authors' explanation (in particular, contrasting with prior explanations that require details of the distillation procedure like logit leakage).

**Reviewer Scores:**

I believe Reviewer bqv2 would have updated their score upon seeing the replication in the additional domain, and that it is possible that Reviewer VLoB would update their score based on the more detailed justifications. Thus, I recommend that this paper be accepted.

---

### Decision · Program_Chairs · 2026-01-26

Accept (Poster)